# Solvent-free processing of lignin into robust room temperature phosphorescent materials

Min Wang[1], Wei-Ming Yin ●[1], Yingxiang Zhai ●[1], Jingyi Zhou[1], Shouxin Liu[1], Jian Li[1], Shujun Li ●[1] ✉, Tony D. James ●[2,3] ✉ & Zhijun Chen ●[1] ✉

Producing room temperature phosphorescent (RTP) materials from biomass resources using a solvent free method is essential but hard to achieve. Here, we discovered that lignin dissolved well in the liquid monomer, 2-hydroxyethyl acrylate (HEA), due to extensive hydrogen bonding and non-bonding interactions between lignin and HEA. Motivated by this discovery, we developed a solvent free system consisting of HEA and urethane dimethacrylate (UDMA) for converting lignin into RTP materials. With this design, lignin generated radicals upon UV irradiation, which initiated the polymerization of HEA (as monomer) and UDMA (as crosslinker). The as-obtained polymer network rigidifies lignin and activates the humidity/water-resistant RTP of lignin with a lifetime of 202.9 ms. Moreover, the afterglow color was successfully tuned to red after loading with RhB via energy transfer (TS-FRET). Using these properties, the as-developed material was used as photocured multiple-emission RTP inks, luminescent coatings and a smart anti-counterfeiting logo for a medicine bottle.

Lignin, the most abundant natural aromatic polymer, is produced as a byproduct in the pulping and paper industry. Where ~60–70 Mt of lignin is produced per year and ~95% is simply burnt[1–5]. As a result, utilizing technical lignin in a value-added manner represents a challenging global hot topic. From a chemical perspective, lignin is created through an enzyme-mediated dehydrogenative polymerization of these units including coumaryl alcohol, coniferyl alcohol and sinapyl alcohol[6,7]. The G, S and H units endow lignin with interesting optical properties, such as, UV blocking, fluorescence, photothermal conversion and room temperature phosphorescence (RTP)[8–14].

Notably, RTP materials exhibit great potential for bioimaging, organic light-emitting diodes, X-ray scintillators and anti-counterfeiting applications[15–26]. Thus, converting lignin to RTP materials is a promising approach for creating value-added applications using lignin. The general method for processing lignin into RTP materials is to rigidify lignin using an external polymer matrix. As such,

the triplet excitons of lignin are protected from quenching by external oxygen and humidity and can migrate to the ground state radiatively, which results in RTP emission. Using this strategy, valuable RTP fibers, films, papers and 3D printable inks can be obtained from lignin[27–33].

Nevertheless, aqueous or ionic liquid solvent is required for processing lignin into RTP materials. Generally, to avoid compromising the RTP performance, aqueous solvents should be removed from the obtained RTP materials via evaporation at the cost of energy and time[10,31,34] (Fig. 1). Notably, ionic liquids can be incorporated with the RTP materials because they do not interfere with the RTP performance. Thus, ionic liquid could be kept in the as-prepared RTP materials unlike other kinds of solvent., but the high price and possible biological toxicity hinders the practical applications of such RTP materials[27,35]. Moreover, RTP emission of the as-obtained materials via solvent processing can be easily quenched by humidity or water. Therefore, the ideal processing of lignin into robust RTP materials requires a solvent free system.

[1]Key Laboratory of Bio-based Material Science & Technology, Northeast Forestry University, Ministry of Education, Harbin, China. [2]Department of Chemistry, University of Bath, Bath, UK. [3]School of Chemistry and Chemical Engineering, Henan Normal University, Xinxiang, China. ✉e-mail: lishujun@nefu.edu.cn; T.D.James@bath.ac.uk; chenzhijun@nefu.edu.cn

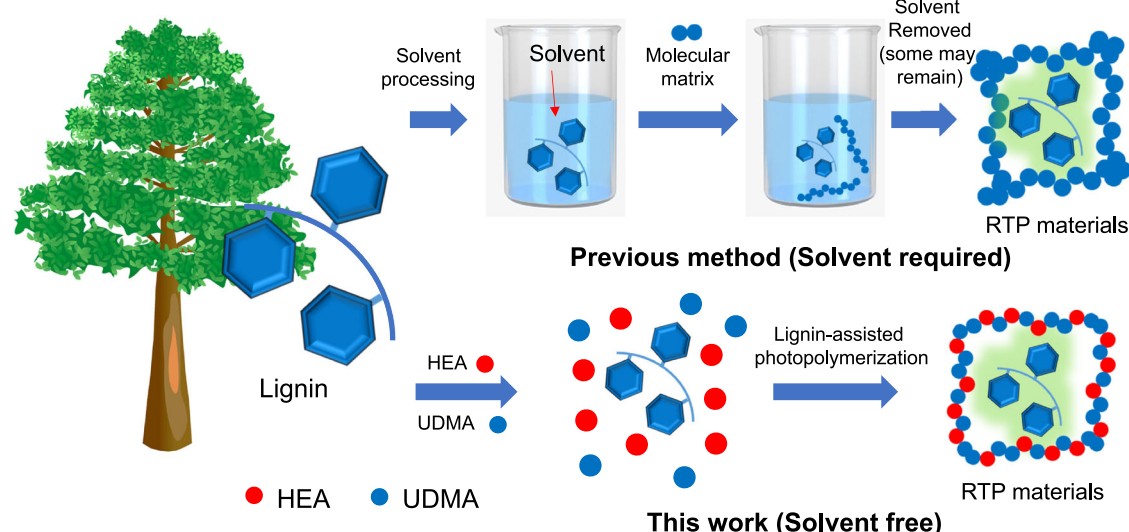

**Fig. 1 | Scheme illustrating the production of RTP materials from lignin.** (Top) RTP materials from lignin via a solvent-processing method and (bottom) our solvent free method.

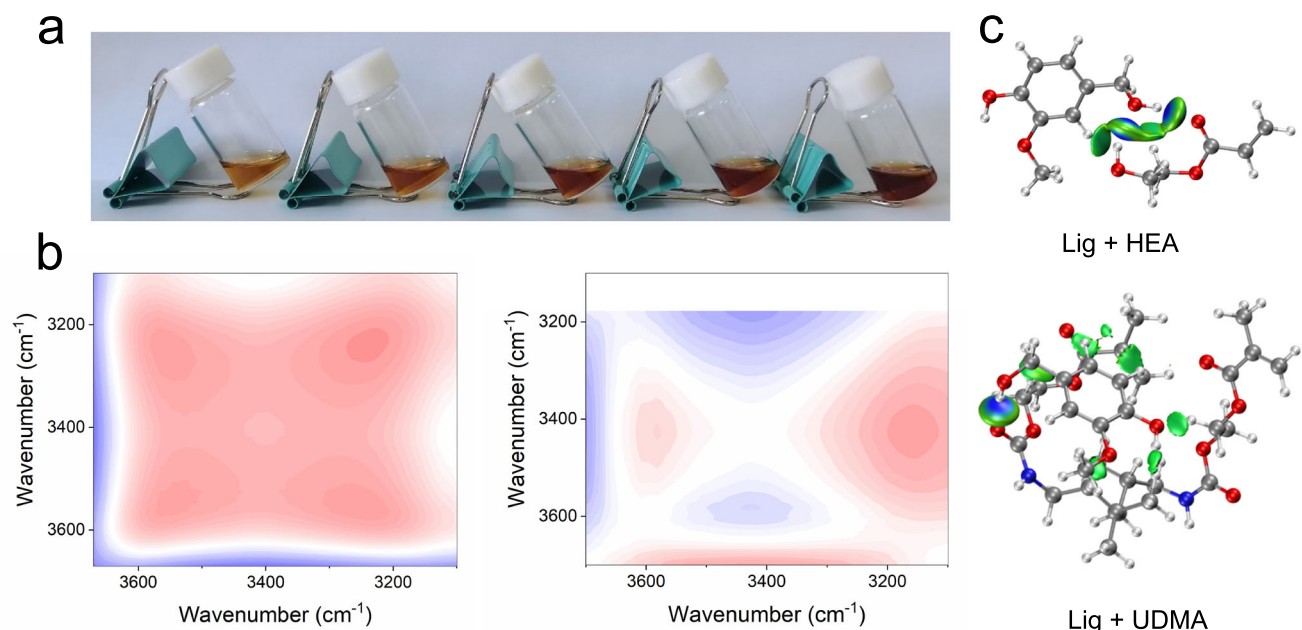

**Fig. 2 | The interaction between lignin and HEA. a** Photographs of homogeneous solution of lignin dissolved in HEA at different concentrations, (the concentrations from left to right were 0.1 wt%, 0.2 wt%, 0.3 wt% and 0.5 wt%). **b** 2D COS synchronous (left) and asynchronous (right) spectra generated from Supplementary Fig. S4. (In 2D COS spectra, the warm color (red) represents positive intensities, while cold color (blue) represents negative intensities). **c** Calculated interactions between lignin and HEA (up) and UDMA (down).

Motivated by this, we developed a solvent-free system consisting of HEA and UDMA for processing lignin into RTP materials (Fig. 1). With this design, lignin dissolved well in the HEA. Thus, no external solvent was required. Lignin generated radicals upon UV irradiation, which initiates the polymerization of HEA (as monomer) and UDMA (as crosslinker). Significantly, the as-obtained polymer network rigidifies lignin and activates RTP.

## Results

### The interaction between lignin and HEA

A homogeneous solution was obtained when the concentration of lignin increased from ~0.1 wt% to ~15 wt% (Fig. 2a and Supplementary Fig. S1) in HEA. To check the generality, a variety of technical lignin including alkaline lignin (AL) and sodium lignosulfonate (SL), were also explored and they both dissolved well in HEA (Supplementary Fig. S2). To understand the interaction, as a control liquid monomer, ethylene glycol and ethyl acrylate, were used. Lignin dissolved well in ethylene glycol but exhibited poor solubility in ethyl acrylate (Supplementary Fig. S3). The result indicated that the -OH group was crucial for dissolving lignin in HEA.

Additionally, FT−IR analysis exhibits intense and broad signals at ~3415 cm⁻¹ (belonging to the O−H stretching vibration) with increased concentration of lignin, indicating increased hydrogen bonding between lignin and HEA (Supplementary Fig. S4)[36]. 2D COS spectra were further generated from Supplementary Fig. S4 to discern the sequential response for different O-H species. According to the Noda's judging rule, both signals at ~3580 cm⁻¹ and ~3210 cm⁻¹ responded earlier than the signal at ~3415 cm⁻¹ according to the synchronous and

asynchronous spectra upon increasing the concentration of lignin (Fig. 2b). These results indicate that the intermolecular hydrogen bonds between lignin and HEA formed and became more intense for concentrated lignin. Significantly, such interactions are particularly beneficial for dissolving lignin in HEA. Theoretical calculations also indicated that lignin interacts strongly with HEA with an intermolecular binding energy value of −60.62 eV. While the intermolecular binding energy value between lignin and UDMA was −49.31 eV, apparently weaker than the former (Fig. 2c and Supplementary Table S1). These calculations agreed well with the experimental results. Since, the experimental results indicated that lignin dissolved well in HEA, but was poorly soluble in UDMA (Supplementary Fig. S5).

To further determine the interaction between lignin and HEA upon UV irradiation, which was crucial for the preparation of RTP materials in the next step, 2D HSQC NMR analysis of the reaction systems consisting of lignin and HEA was conducted before and after UV irradiation (Supplementary Fig. S6). Both the typical signals of lignin and polymerized HEA (PHEA) were observed from 2D HSQC NMR spectra, suggesting the polymerization of HEA occurred in the presence of lignin after UV irradiation. To further understand the reaction, $^1$H NMR analysis of the reaction system was then conducted before and after UV irradiation (Supplementary Fig. S7). The result indicated that the signals of the double bond was decreased, indicating that the polymerization of HEA had occurred. Moreover, a new signal, assigned as H of −O−CH$_2$− (located at 5.3 ppm), appeared. Meanwhile, the $^{31}$P NMR suggested that the signals of phenolic hydroxyl moieties from lignin almost disappeared after the reaction (Supplementary Fig. S8). This result suggested that phenolic moieties of lignin reacted with the double bond of HEA during the reaction and polymerization of HEA. Further FT-IR analysis also confirmed the reaction. Signals at 1242 cm$^{-1}$, assigned as −O−CH$_2$− increased after reaction (Supplementary Fig. S9)[37]. Additionally, the GPC trace also confirmed that the as-obtained products exhibited higher molecular weight than raw lignin, indicating chemical reaction had occurred between the lignin and the polymerized HEA (Supplementary Fig. S10). All these results indicated that HEA could be polymerized and linked with lignin via −O−CH$_2$− bonds upon UV irradiation.

## Preparation and RTP properties of Lig-Poly

To obtain RTP materials, lignin was dissolved in a mixture of HEA and UDMA with a ratio of 0.1 wt% and 3:7 (w/w). Upon UV irradiation the generated OH photoradicals of lignin, as determined by ESR spectroscopy, ~80% double bond conversion was achieved (Fig. 3a and Supplementary Fig. S11)[38,39]. This method was then used to prepare Lig-Poly. The mechanical performance of Lig-Poly was then investigated. Lig-Poly exhibited a tensile strength of 52.8 MPa and elongation at break of 9.8% (Supplementary Fig. S12). As a comparison, cured samples (Poly) in the absence of lignin exhibited 34.7 MPa and 9.4%. These results indicate that lignin is beneficial for the mechanical enhancement of the polymers. Further dynamic mechanical analysis (DMA) of Lig-Poly indicated that the glass transition temperature was 68 °C, confirming that it can be thermally processed when the temperature is above 68 °C (Supplementary Fig. S13). As a control, no photocuring was observed when DMPO a radical scavenger was added into the reaction mixtures (Supplementary Fig. S14). Also, the formula in the absence of lignin was not cured upon UV excitation (Supplementary Fig. S15). All these results confirm photoradical-induced polymerization catalyzed by lignin. After that, the mechanical performance of the cured sample was measured. The hardness and Young's modulus of the sample were 203.57 MPa and 3.93 GPa, respectively (Supplementary Fig. S16).

Interestingly, exposure of the precursors to UV light sources led to an enhancement of both the RTP intensity and lifetime (Fig. 3b and Supplementary Fig. S17). After 30 min, the RTP performance of Lig-Poly reached equilibrium and exhibited long-lasting emission for 202.9 ms (Supplementary Fig. S18). As a control, only very weak RTP emission, attributed to the clustering-induced emission, was observed for the Poly sample in the absence of lignin (Supplementary Fig. S19). This result illustrates that lignin was the main chromophore of Lig-Poly. To understand the RTP emission from Lig-Poly, theoretical calculations were conducted. The result indicted that lignin exhibits stronger interactions with the polymerized network than the monomers (Supplementary Fig. S20 and Supplementary Table S1). Such interactions restrict the molecular motion, resulting in RTP emission[40–42]. Additionally, mechanical analysis indicated that increasing the exposure time of the formula with UV light sources led to an enhanced tensile strengthen and reduced strain ratios (Supplementary Fig. S21). This result indicates that the rigidity of the as-formed Lig-Poly increased with extended photocuring. Thus, exposure of the precursors to UV light sources led to an increased polymerization degree. The enhanced polymerization degree induced a higher crosslinking density, contributing to a rigidified environment and promotion of the RTP performance of Lig-Poly[43,44]. This helps explain the enhanced intensity and lifetime of Lig-Poly with an extended curing time. Notably, the RTP performance of Lig-Poly was dependent on the concentration of lignin. Increasing the concentration of lignin from 0.01% to 0.1% enhanced the RTP performance (Supplementary Fig. S22). However, further increasing the concentration of lignin decreased the RTP emission and lifetime because of the π–π stacking of aromatic units in lignin. Additionally, the samples were prepared using different types of lignin including sodium lignosulfonate (SL) and alkaline lignin (AL) and all these materials exhibited RTP emission, suggesting the generalization of the method (Supplementary Fig. S23). Moreover, all the samples prepared using different crosslinkers and HEA in the presence of lignin exhibited RTP emission, further indicating the generalization of the strategy (Supplementary Fig. S24).

Lig-Poly exhibited humidity/water-resistant RTP emission, which has rarely been observed for similar systems. Specifically, the RTP lifetime did not decrease when it was treated with 90% humidity or immersed in water for 24 h (Fig. 3c). To further understand the phenomenon, the surface tension and contact angle of Lig-Poly were measured (Supplementary Fig. S25). The values of surface tension and contact angle were 65.53 mN/m and 41.6°, respectively. Interestingly, the surface of Lig-Poly exhibited hydrophilicity rather than hydrophobicity. This was because the water molecules form hydrogen bonds with the surface hydroxyl moieties from the polymerized HEA of Lig-Poly[45]. Such interactions induced formation of dense molecular networks preventing the water entering the network. As a result, the triplet excitons were well protected in the Lig-Poly and RTP emission was stable when the sample was immersed into water.

The stability of Lig-Poly was further investigated in organic solvents. The results indicated that the RTP lifetime of Lig-Poly was not compromised when it was immersed in different organic solvents, such as, ethanol, methanol, ethyl acetate, tetrahydrofuran, dichloromethane and acetonitrile for 24 h (Fig. 3d). However, the RTP of Lig-Poly was sensitive to temperature. When the temperature decreased, both the intensity and lifetime of Lig-Poly increased, attributed to enhanced radiative migration of triplet excitons (Supplementary Fig. S26).

Notably, spectral analysis revealed a significant overlap between the RTP emission band of Lig-Poly and the absorption spectrum of RhB (Supplementary Fig. S27), suggesting the potential for efficient energy transfer between these two components. As such we successfully incorporated RhB into the Lig-Poly matrix to construct a novel Lig-Poly/RhB composite system. Remarkably, the resulting Lig-Poly/RhB complex demonstrated distinct red afterglow emission properties, with a measured lifetime of 167.7 ms, as clearly illustrated in Fig. 3e and f. To further understand the energy transfer, the lifetime of Lig-Poly before and after loading with RhB was measured. Time-resolved photoluminescence analysis of lignin phosphorescence at 500 nm (λ$_{ex}$ = 320 nm) revealed a

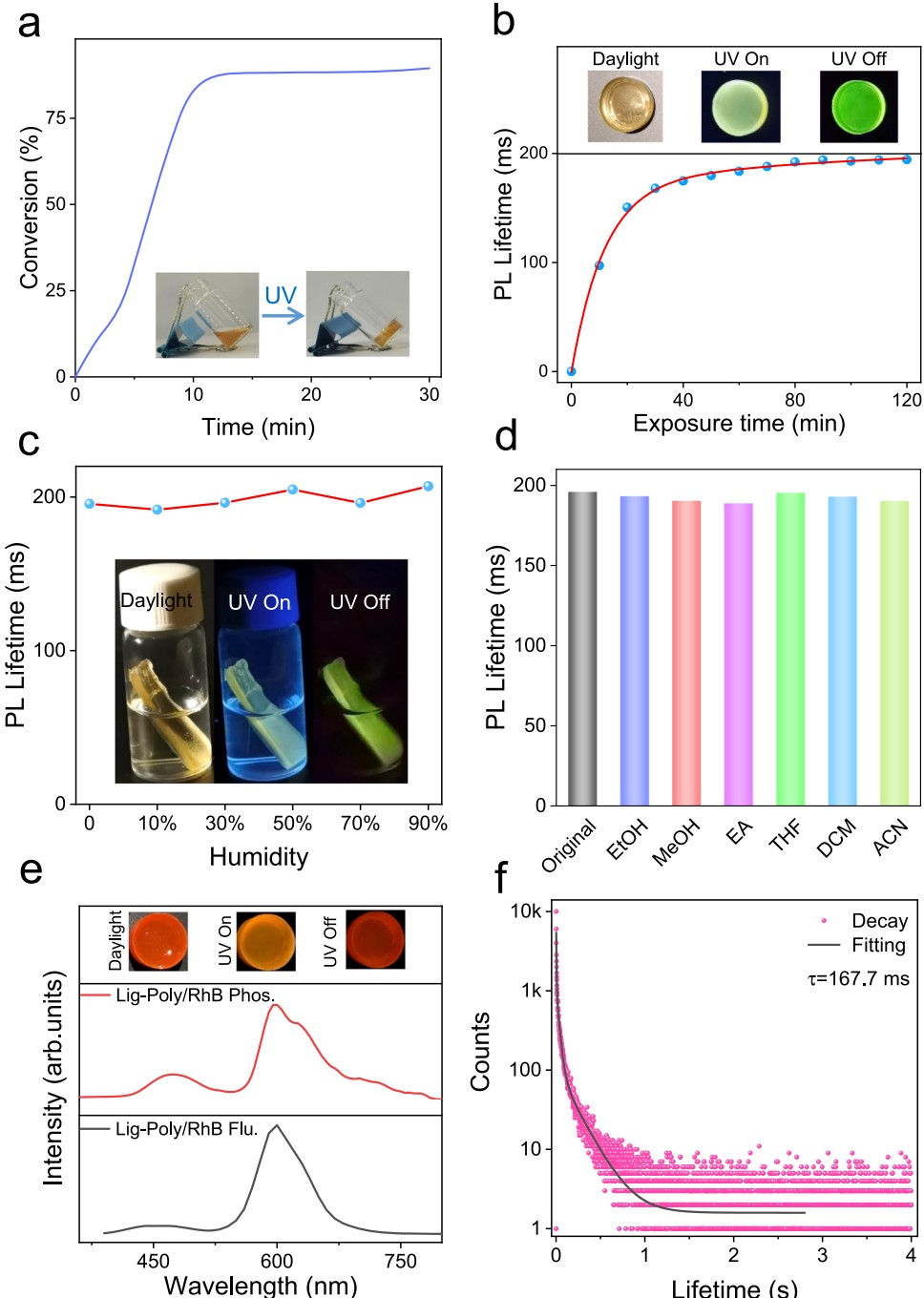

**Fig. 3 | Preparation and RTP properties of Lig-Poly. a** Double bond conversion of Lig-Poly during polymerization, inset: images of Lig-Poly before (left) and after (right) irradiation with UV light. **b** RTP lifetimes of Lig-Poly with different irradiation times during polymerization, inset: images of Lig-Poly in daylight (left), under excitation of UV light (middle) and after turning off the UV light (right). **c** RTP lifetimes of Lig-Poly under different humidity (red line), inset: images of Lig-Poly soaking in water in daylight (left), under the excitation of UV light (middle) and after turning off the UV light (right). (Conditions for the images: The samples were immersed into water for 24 h. After that, the digital images of the sample were taken under bright field, UV field and after switching off the UV light sources). **d** RTP lifetimes of Lig-Poly before (gray) and after (colored) when soaked in different solvents. **e** Fluorescence (black line) and RTP emission (red line) spectra of Lig-Poly/RhB excited by 320 nm light, inset: images of Lig-Poly/RhB in daylight (left), under excitation of UV light (middle) and after turning off the UV light (right). **f** RTP decay curve of Lig-Poly/RhB.

significant reduction in the average lifetime from 203.2 ms to 141.5 ms upon incorporation of RhB into Lig-Poly (Supplementary Fig. S28). This lifetime reduction provides compelling evidence for non-radiative Förster resonance energy transfer (FRET) from the triplet state of lignin donors to the singlet state of RhB acceptors in the Lig-Poly/RhB system, specifically through a triplet-to-singlet FRET (TS-FRET) mechanism. The observed lifetime change effectively rules out the possibility of a simple emission-reabsorption process, since for that the donor lifetime would

not change, which is consistent with previous reports[31,46,47]. Further evidence supporting the TS-FRET mechanism was obtained through selective excitation experiments. When Lig-Poly/RhB was excited at 500 nm (acceptor excitation), no emission was detected in the time-gated spectra. In contrast, distinct emission was observed upon excitation at 320 nm (donor excitation) (Supplementary Fig. S28). These results unequivocally confirm that the long-lived triplet excitons of lignin donors serve as the exclusive source for populating the singlet

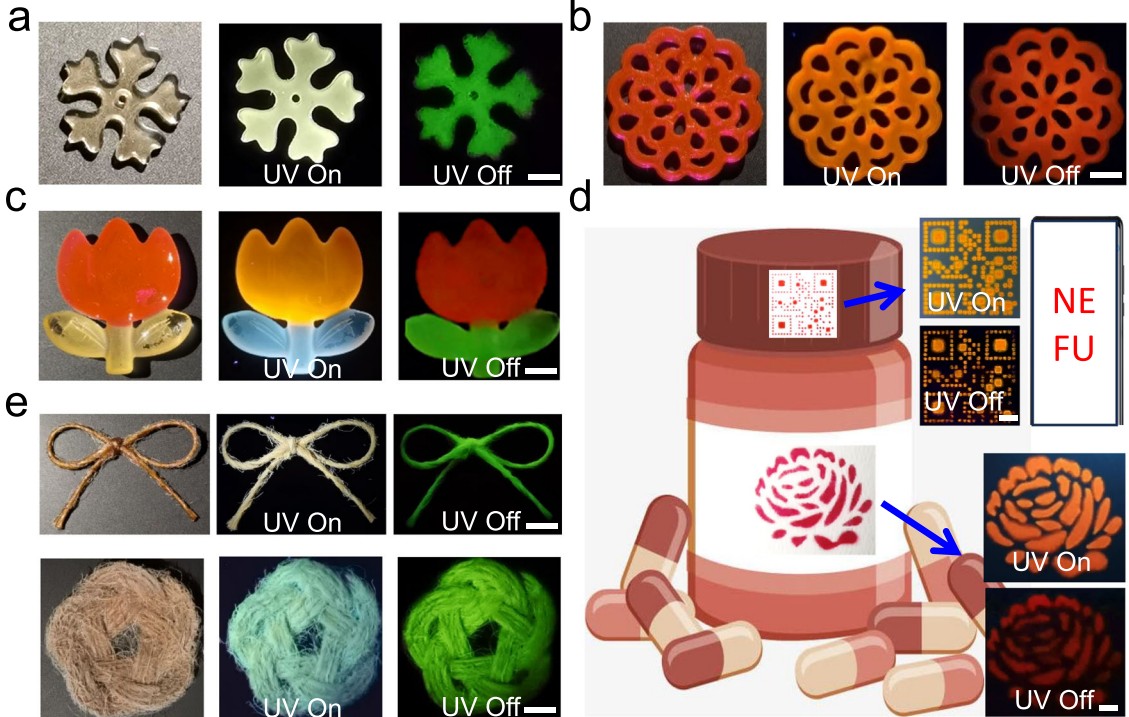

**Fig. 4 | Application of Lig-Poly and Lig-Poly/RhB. a** Images of Lig-Poly in daylight (left), Lig-Poly upon excitation of UV light (middle) and Lig-Poly after switching off the UV light (right) (scale bar = 5 mm). **b** Images of Lig-Poly/RhB in daylight (left), Lig-Poly/RhB upon excitation of UV light (middle) and Lig-Poly/RhB after switching off the UV light (right) (scale bar = 5 mm). **c** Images of Lig-Poly and Lig-Poly/RhB in daylight (left), Lig-Poly and Lig-Poly/RhB upon excitation of UV light (middle) and Lig-Poly and Lig-Poly/RhB after switching off the UV light (right) (scale bar = 5 mm). **d** Image of a medicine bottle with 2D code on the cap and peony made for the bottle (scale bar = 3 mm). **e** Images of single twine treated by Lig-Poly and knitting produced using twine treated with Lig-Poly (scale bar = 5 mm).

state of RhB acceptors through an efficient TS-FRET process, generating persistent delayed fluorescence. Notably, the energy transfer efficiency between lignin and RhB in the Lig-Poly matrix exhibited a strong dependence on RhB concentration, suggesting the potential for tunable photophysical properties in this system. The energy transfer efficiency increased from 0% to 30.4% when the concentration ratio of RhB to lignin increased from 0:1 to 0.04:1 (w/w) (Supplementary Table S2).

## Applications

Using the photocuring of Lig-Poly different shapes were easily obtained using a template and the in situ exposure of the precursors to UV light sources (Fig. 4a). In addition, 3D materials with red phosphorescence were prepared by exposure of the precursors of Lig-Poly/RhB to UV light (Fig. 4b). Additionally, Lig-Poly and Lig-Poly/RhB were easily combined to generate shapes with two colors of phosphorescence (Fig. 4c). Attributed to the inherent properties of the photocured materials, Lig-Poly/RhB can also be used for anti-counterfeiting coatings (Fig. 4d). To further demonstrate the potential, Lig-Poly/RhB was processed into a QR code and tag, which can be used on a medicine bottle for anti-counterfeiting purposes. All these systems mentioned above-exhibited afterglow RTP emission after removing the excitation sources. Moreover, the Lig-Poly was used as a photo-cured coating for twine (Fig. 4e). The obtained RTP twines were then processed into different shapes. Using thermal processing, Lig-Poly and Lig-Poly/RhB were easily processed into different shapes with RTP emission (Supplementary Fig. S29). Interestingly, Lig-Poly and Lig-Poly/RhB returned to their initial shapes with RTP emission due to inherent shape-memory properties. The whole process was reversible several times. The effect of thermal processing on RTP performance of Lig-Poly and Lig-Poly/RhB was quantitively determined. Significantly, both the RTP intensity and lifetime of Lig-Poly and Lig-Poly/RhB were not compromised after thermal processing. Moreover, they did not

significantly change after being recycled 5 times by thermal processing (Supplementary Fig. S30).

## Discussion

In summary, we have developed a solvent-free RTP system consisting of lignin, HEA and UDMA. The strong hydrogen bonding interaction between HEA and lignin facilitates the dissolution of lignin in a mixture of HEA and UDMA. Then, dissolved lignin generated photoradicals, triggering the polymerization of HEA and UDMA. The as-formed rigid polymer network confined lignin and triggered green RTP emission from lignin with a lifetime of 202.9 ms. Moreover, the afterglow color was successfully converted to red after loading with RhB using a TS-FRET strategy. Using these properties, the solvent-free RTP system was used to produce functional luminescent materials for coatings and anti-counterfeiting applications. Considering the sustainability, convenience, inexpensive and energy saving properties, this research will enable the synthesis of practical RTP materials from renewable resources on a large scale. Moreover, considering the simplicity in fabrication, high transparency and hardness, the as-obtained Lig-Poly exhibited potential to be used as RTP glass, which has great potential in optical displays, wearable devices, and portable optoelectronics[48].

## Methods
### Preparation of Lig-Poly
One milligram of EL (SL or AL) was added in 300 mg HEA and dissolved completely by assistance of ultrasonic, and then 700 mg UDMA (TEGDA, TPGDA or TMPTA) was added into the solution to obtain the homogeneous solution by stirring and ultrasound. After that, the solution was put into mold and exposed under UV light sources (365 nm, 70 mW cm⁻²) for 20 min for photocuring.

## Preparation of Poly

The control sample was prepared as following: 1 mg of AIBN was added in 300 mg HEA and dissolved completely by assistance of ultrasonic, and then 700 mg UDMA was added into the solution to obtain the homogeneous solution by stirring and ultrasound. After that, the solution was put into mold and exposed under UV light sources (365 nm, 70 mW cm$^{-2}$) for 20 min for photocuring.

## Data availability

All relevant data are included in this article and its Supplementary Information file. All data are available from the corresponding author (Zhijun Chen) upon request. Source data are provided with this paper.

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

## Acknowledgements

Z.J.C. wishes to thank the National Natural Science Foundation of China (32471803) and Fundamental Research Funds for the Central Universities (2572022CG02). T.D.J. wishes to thank the University of Bath and the Open Research Fund of the School of Chemistry and Chemical Engineering, Henan Normal University (2020ZD01) for support.

## Author contributions

Conceptualization: Z.C., S. Li., and T.D.J.; Methodology: M.W. and W.Y.; Investigation: M.W., Y.Z., and J.Z.; Visualization: M.W., S. Liu., J.L., and S. Li.; Supervision: Z.C., S. Li., and T.D.J.; Writing-original draft: All authors; Writing-review and editing: All authors.

## Competing interests

The authors declare no competing interests.
