## [Transparent Peer Review file · Nature Communications]

Solvent-free processing of lignin into robust room temperature phosphorescent materials

Corresponding Author: Dr Tony James

Version 0:

Reviewer comments:

Reviewer #1

(Remarks to the Author)

The authors produced room temperature phosphorescent (RTP) materials from biomass resources using a solvent free method. In the work, they discovered that lignin dissolved well in the liquid monomer, 2-hydroxyethyl acrylate (HEA), due to extensive hydrogen bonding and non-bonding interactions between lignin and HEA. Motivated by this discovery, they developed a solvent free system consisting of HEA and urethane dimethacrylate (UDMA) for converting lignin into RTP materials. With this design, lignin generated radicals upon UV irradiation, which initiated the polymerization of HEA (as monomer) and UDMA (as crosslinker). The as-obtained polymer network rigidifies lignin and activates the humidity/water-resistant RTP of lignin with a lifetime of 202.9 ms. Moreover, the afterglow color was successfully tuned to red after loading with RhB via energy transfer (TS-FRET). Generally, the manuscript was well written and organized. I recommended the publication after the authors addressed these minor problems.

1. The author should compare the RTP performance of lignin at different concentration in the system since it is important to understand the RTP behavior of lignin?
2. The authors showed the generality of lignin dissolved in the HEA. Did all these lignin exhibited RTP if they were trapped in the polymerized matrix?
3. How about the mechanical performance, such as hardness, Young's modulus of the cured samples?
4. For me, the cured samples are like glasses. RTP glasses were very promising for wide applications. The authors should add some discussion about this aspect.
5. The enqueued RTP in Figure 3c inset was impressive. The authors should describe more details about this figure.

Reviewer #2

(Remarks to the Author)

Chen et al reported to prepare RTP materials from lignin in a solvent free manner. As-obtained materials exhibit robust and humidity-independent RTP emission. The concept demonstrated in the work is novel and interesting. Thus, I recommend the publication after these minor problems were addressed.

1. The authors should discuss more advantages of this system, particularly the priorities to the ionic liquid systems.
2. Did the authors consider how the concentration of lignin affected the emission? It is better to add this result.
3. The authors clearly demonstrated the importance of HEA. Did the materials still exhibit RTP when HEA maintained and other crosslinker reagents were used?
4. The authors should explain why exposure of the precursors to UV light sources led to an enhancement of both the RTP intensity and lifetime.
5. What is the energy transfer efficiency between lignin and RhB?

Reviewer #3

(Remarks to the Author)

interesting work and outstanding outcomes. However, the analysis lacks basic fundamentals required for lignin insights. I

would suggest that the authors conduct the following and provide these in the revised paper
1- P NMR, and HSQC NMR of the lignin before and after the reaction and some fundamental discussion on how the structures of lignin is changed during its reaction with HEA.
2- surface tension and contact angle to verify the hydrophobicity and wmoisture interaction.
3- I could not find the methodology well detailed.

Version 1:

Reviewer comments:

Reviewer #1

(Remarks to the Author)

The manuscript can be accepted as the authors have adressed the questions.

Reviewer #2

(Remarks to the Author)

All the raised points have been well addressed. It can be accepted.

Reviewer #3

(Remarks to the Author)

The authors address the comments appropriately, the paper can be accepted for publication now.

Response

REVIEWER COMMENTS

Reviewer #1 (Remarks to the Author):

The authors produced room temperature phosphorescent (RTP) materials from biomass resources using a solvent free method. In the work, they discovered that lignin dissolved well in the liquid monomer, 2-hydroxyethyl acrylate (HEA), due to extensive hydrogen bonding and non-bonding interactions between lignin and HEA. Motivated by this discovery, they developed a solvent free system consisting of HEA and urethane dimethacrylate (UDMA) for converting lignin into RTP materials. With this design, lignin generated radicals upon UV irradiation, which initiated the polymerization of HEA (as monomer) and UDMA (as crosslinker). The as-obtained polymer network rigidifies lignin and activates the humidity/water-resistant RTP of lignin with a lifetime of 202.9 ms. Moreover, the afterglow color was successfully tuned to red after loading with RhB via energy transfer (TS-FRET). Generally, the manuscript was well written and organized. I recommended the publication after the authors addressed these minor problems.

1. The author should compare the RTP performance of lignin at different concentration in the system since it is important to understand the RTP behavior of lignin?

A: Thanks for the comments.

We measured the RTP performance of lignin at different concentrations. Increasing the concentration of lignin from 0.01% to 0.1% enhanced the RTP performance. However, further increasing the concentration of lignin decreased the RTP emission and lifetime because of the π - π stacking of the aromatic units in lignin.

We added the result as **Supplementary Fig. S22**.

Supplementary Fig. S22 Emissions (a) and RTP lifetimes (b) of Lig-Poly made using different amounts of lignin. In all cases $\lambda_{exc.} = 320$ nm, $\lambda_{collected} = 500$ nm, delay time = 10 ms.

2. The authors showed the generality of lignin dissolved in the HEA. Did all these lignin exhibited RTP if they were trapped in the polymerized matrix?

A: Thanks for the comments.

We prepared the samples using different types of lignin. All of these materials exhibited RTP emission.

We added the result as **Supplementary Fig. S23**.

Supplementary Fig. S23 RTP emission spectra and lifetimes of **Lig-Poly** prepared using different kinds of lignin. Standard (black line) and delayed (red line) emission spectra of a) SL-Poly (sodium lignosulfonate) and b) AL-Poly (alkaline lignin), Inset: the images of polymers in daylight (left), upon excitation by UV light source (middle) and after switching off the UV light source (right); Phosphorescence lifetime of c) SL-Poly and d) AL-Poly. In all cases $\lambda_{exc.} = 320 \text{ nm}$, $\lambda_{collected} = 500 \text{ nm}$, delay time = 10 ms.

3. How about the mechanical performance, such as hardness, Young's modulus of the cured samples?

A: Thanks for the comments.

We measured the mechanical performance of the cured samples including hardness and Young's modulus.

We added the result as **Supplementary Fig. S16**.

Supplementary Fig. S16 Mechanical properties of Lig-Poly.

4. For me, the cured samples are like glasses. RTP glasses were very promising for wide applications. The authors should add some discussion about this aspect.

A: Thanks for the comments.

“Considering of the simplicity in fabrication, high transparency and hardness, the as-obtained Lig-Poly exhibited potential to be used as RTP glass, which has great potential in optical displays, wearable devices, and portable optoelectronics (Nat. Commun., 2024, **15**, 9491).

We added the content in the revised manuscript.

5. The enqueued RTP in Figure 3c inset was impressive. The authors should describe more details about this figure.

A: Thanks for the comments.

The samples were immersed into water for 24 h. After that, the digital images of the sample were taken under bright field, UV field and after switching off the UV light sources.

We added the content in the revised manuscript.

Reviewer #2 (Remarks to the Author):

Chen et al reported to prepare RTP materials from lignin in a solvent free manner. As-obtained materials exhibit robust and humidity-independent RTP emission. The concept demonstrated in the work is novel and interesting. Thus, i recommend the publication after these minor problems were addressed.

1. The authors should discuss more advantages of this system, particularly the priorities to the ionic liquid systems.

A: Thanks for the comments.

Producing RTP materials from natural resources is crucial for sustainability. Amongst these natural resources, lignin is particularly welcome because of the unique aromatic units, abundance and low cost. Moreover, these resources are regarded as waste materials and the value has been under evaluated. Thus, converting lignin into RTP materials is not only vital for the sustainability of RTP materials, it is also important for generating added-value applications of lignin.

In the previous report, lignin can only be converted into RTP materials assisted by solvents. Amongst these solvents, ionic solvents dissolved lignin very well. Additionally, ionic liquids do not

interfere with the RTP performance of the prepared materials. As such, ionic liquids can remain in the as-prepared RTP materials unlike other kinds of solvent.

However, ionic liquids are expensive. Additionally, the RTP emission of such materials is easily quenched by humidity or water due to the hydrophilicity of ionic liquids.

To conquer such challenges, we developed a solvent free system for producing robust and water-proof RTP materials from lignin.

We added this content in the revised manuscript.

2. Did the authors consider how the concentration of lignin affected the emission? It is better to add this result.

A: Thanks for the comments.

We measured the RTP performance of lignin at different concentration. Increasing the concentration of lignin from 0.01% to 0.1% enhanced the RTP performance. However, further increasing the concentration of lignin decreased the RTP emission and lifetime because of the π - π stacking of the aromatic units in lignin.

We added the result as **Supplementary Fig. S22**.

Supplementary Fig. S22 Emissions (a) and RTP lifetimes (b) of Lig-Poly made using different contents of lignin. In all cases $\lambda_{exc.}=320$ nm, $\lambda_{collected}=500$ nm, delay time = 10 ms.

3. The authors clearly demonstrated the importance of HEA. Did the materials still exhibit RTP when HEA maintained and other crosslinker reagents were used?

A: Thanks for the comments.

We prepared the samples using different crosslinkers and HEA in the presence of lignin. All of these materials exhibited RTP emission.

We added the result as **Supplementary Fig. S24**.

Supplementary Fig. S24 Fluorescence and phosphorescence emission spectra and RTP lifetimes of materials made using other crosslinkers. Fluorescence (black line) and phosphorescence (red line) emission spectra of a) Lig-Poly-TEGDA made with crosslinker TEGDA; b) Lig-Poly-TPGDA made with crosslinker TPGDA; c) Lig-Poly-TMPTA made with crosslinker TMPTA; d) RTP lifetimes of materials made using these three crosslinkers. In all cases $\lambda_{exc.} = 320$ nm, $\lambda_{collected} = 500$ nm, delay time = 10 ms.

4. The authors should explain why exposure of the precursors to UV light sources led to an enhancement of both the RTP intensity and lifetime.

A: Thanks for the comments.

Exposure of the precursors to UV sources led to an increased polymerization degree. The enhanced polymerization induced a higher crosslinking density, contributing to a rigidified environment and promotion of the RTP performance of the **Lig-Poly**.

We added this content in the revised manuscript.

5. What is the energy transfer efficiency between lignin and RhB?

A: Thanks for the comments.

The energy transfer efficiency between lignin and RhB in **Lig-Poly** was dependent on the concentration of RhB. The energy transfer efficiency increased from 0% to 30.4% when the

concentration of RhB to lignin increased from 0:1 to 0.04:1 (w/w).

We added the result as **Supplementary Table S2**.

Supplementary Table S2 Summary of energy transfer (Φ_{et}) efficiency.

Acceptor	Donor (Lignin) and acceptor (RhB) Ratio (w:w)	Average lifetime (ms) of Lig-Poly/RhB at 500 nm ($\lambda_{ex} = 320$ nm)	Energy Transfer Efficiency (%)
/	1:0	203.2	0
RhB	1:0.01	172.0	15.4
RhB	1:0.02	165.9	18.4
RhB	1:0.03	158.5	22.0
RhB	1:0.04	141.5	30.4

Reviewer #3 (Remarks to the Author):

interesting work and outstanding outcomes. However, the analysis lacks basic fundamentals required for lignin insights. I would suggest that the authors conduct the following and provide these in the revised paper

- 1- ³¹P NMR, and HSQC NMR of the lignin before and after the reaction and some fundamental discussion on how the structures of lignin is changed during its reaction with HEA.

A: Thanks for the comments.

To further determine the interaction between lignin and HEA upon UV irradiation, which was crucial for the preparation of RTP materials in the next step, 2D HSQC NMR analysis of the reaction systems consisting of lignin and HEA was conducted before and after UV irradiation (**Supplementary Fig. S6**). Both the typical signals of lignin and polymerized HEA were observed from 2D HSQC NMR spectra, suggesting the polymerization of HEA occurred in the presence of lignin after UV irradiation. To further understand the reaction, ¹H NMR analysis of the reaction system was then conducted before and after UV irradiation (**Supplementary Fig. S7**). The result indicated that the signals of the double bond was decreased, indicating that the polymerization of HEA had occurred. Moreover, a new signal, assigned as H of -O-CH₂- (located at 5.3 ppm), appeared. Meanwhile, the ³¹P NMR suggested that the signals of phenolic hydroxyl moieties from lignin almost disappeared after the reaction (**Supplementary Fig. S8**). This result suggested that phenolic moieties of lignin reacted with the double bond of HEA during the reaction and polymerization of HEA (ACS Sustainable Chem. Eng. 2018, 6, 337-348). Further FT-IR analysis also confirmed the reaction. Signals at 1242 cm⁻¹, assigned as -O-CH₂- increased after reaction (**Supplementary Fig. S9**). Additionally, the GPC trace also confirmed that the as-obtained products exhibited higher molecular weight than raw lignin, indicating chemical reaction had occurred between the lignin and the polymerized HEA (**Supplementary Fig. S10**). All these results indicated that HEA could be polymerized and linked with lignin via -O-CH₂- bonds upon UV irradiation.

We added the results as **Supplementary Fig. S6 ~ S10**.

Supplementary Fig. S6 2D HSQC of the mixtures consisting of lignin (15 mg) and HEA (85 mg) in DMSO- d_6 (0.7 mL) before (b) and after (d) UV irradiation for 6 h.

Supplementary Fig. S7 ^1H NMR of lignin (a) and the mixtures consisting of lignin (7.5 mg) and HEA (42.5 mg) in DMSO-d_6 (0.7 mL) before (b) and after (c) UV irradiation for 6 h.

Supplementary Fig. S8 ^{31}P NMR spectra of the mixtures consisting of lignin (7.5 mg) and HEA (42.5 mg) before and after UV irradiation for 6 h.

Supplementary Fig. S9 FT-IR spectra of the mixtures consisting of lignin (7.5 mg) and HEA (42.5 mg) before and after UV irradiation for 6 h.

Supplementary Fig. S10 GPC trace of the raw lignin and the mixtures consisting of lignin (7.5 mg) and HEA (42.5 mg) after UV irradiation for 6 h.

2- surface tension and contact angle to verify the hydrophobicity and moisture interaction.

A: Thanks for the comments.

We measured the surface tension and contact angle for **Lig-Poly**. The value of surface tension and contact angle was 65.53 mN/m and 41.6°, respectively.

Interestingly, the surface of **Lig-Poly** exhibited hydrophilicity rather than hydrophobicity. This was because the water molecules form hydrogen bonds with the surface hydroxyl moieties from the polymerized HEA of **Lig-Poly** (Chin. J. Chem. 2023, 41, 2289-2295). Such interactions resulted in the formation of a dense molecular network and prevented water from entering the network. As a result, the triplet excitons were protected in **Lig-Poly** and RTP emission was stable when the sample was immersed into water.

We added the result as **Supplementary Fig. S25**.

Supplementary Fig. S25 The water contact angle of cured **Lig-Poly**.

3- I could not find the methodology well detailed.

A: Thanks for the comments.

We added detailed method.

Preparation of Lig-Poly

1 mg of EL (SL or AL) was added to 300 mg of HEA and dissolved completely using ultrasound treatment, and then 700 mg UDMA (TEGDA, TPGDA or TMPTA) was added into the solution to

obtain a homogeneous solution by stirring and ultrasound treatment. After that, the solution was placed into a mold and exposed to UV light sources (365 nm, 70 mW cm⁻²) for 20 min for photocuring.

Preparation of Poly

The control sample was prepared as follows: 1 mg of AIBN was added to 300 mg HEA and dissolved completely using ultrasound treatment, and then 700 mg UDMA was added into the solution to obtain the homogeneous solution by stirring and ultrasound. After that, the solution was placed into a mold and exposed to UV light sources (365 nm, 70 mW cm⁻²) for 20 min for photocuring.